# IFG: Internet-Scale Guidance for Functional Grasping Generation

Anonymous Review

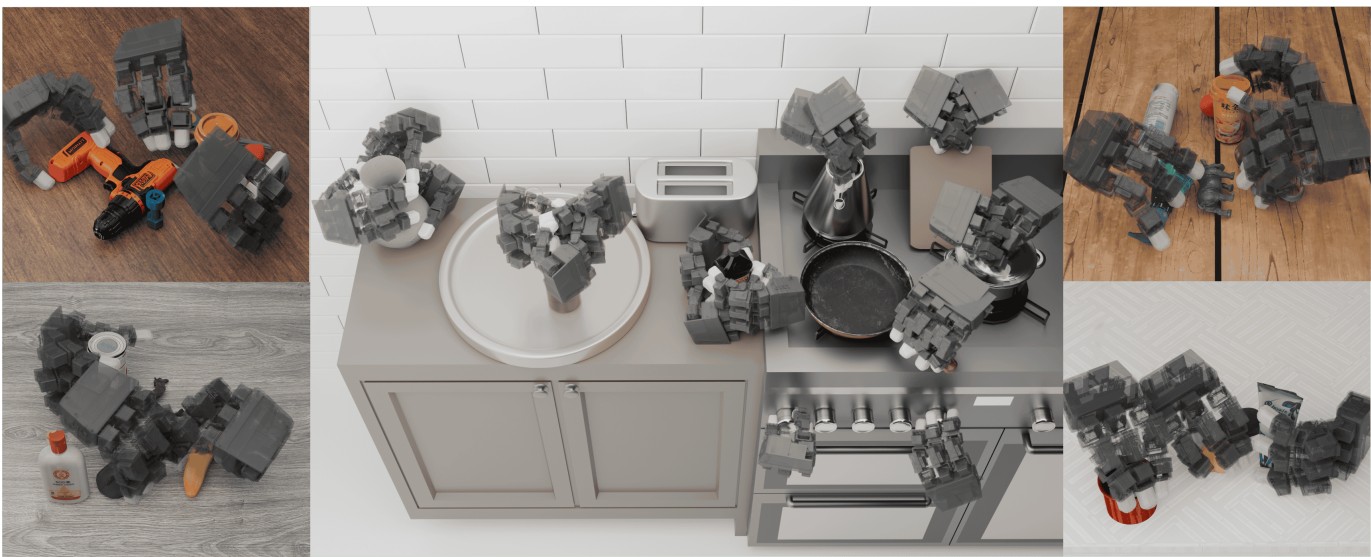

Fig. 1: IFG enables the generation of dexterous, functional grasps in cluttered, realistic scenes. It first uses a vision-language model to identify task-relevant regions on objects, then uses geometrically precise force closure in simulation to ground the finger joints. The resulting dataset, and the diffusion model trained on it, encode both semantic and geometric understanding of the scene without any hand-collected data.

*Abstract*—Large Vision Models trained on internet-scale data have demonstrated strong capabilities in segmenting and semantically understanding object parts, even in cluttered scenes. However, while these models can direct a robot toward the general region of an object, they lack the geometric understanding required to precisely control dexterous robotic hands for 3D grasping. To overcome this, our key insight is to leverage simulation with a force-closure grasping generation pipeline that understands local geometries of the hand and object in the scene. Because this pipeline is slow and requires ground-truth observations, the generated dataset is distilled into a diffusion model that can operate on camera point clouds. By combining the global semantic understanding of internet-scale models with the geometric precision of a simulation-based locally-aware force-closure, IFG achieves high-performance semantic grasping without any manually collected training data. For visualizations, please visit our website at https://ifgrasping.github.io/

## I. INTRODUCTION

Recent advances in vision-language models (VLMs) have led to impressive results across a range of perception tasks, including image captioning, visual question answering, and open-world object recognition. Trained on large-scale datasets pairing images with natural language, these models exhibit a strong ability to align visual and linguistic information, enabling semantic understanding that generalizes across diverse contexts. This success has inspired interest in leveraging VLMs for robotics applications such as instruction following, semantic goal specification, and high-level planning. While these initial applications show promise, significant limitations remain. Most notably, current VLMs lack a grounded understanding of physical space—they cannot reliably reason about 3D geometry,

spatial relationships, or the dynamics of physical interaction. Consequently, they struggle with planning or executing precise motor actions in the real world. Although VLMs can identify visual content, they do not inherently understand how to interact with it. Addressing this disconnect between perception and control is a major challenge in robotic grasping systems.

We seek an approach that avoids manual data collection through means like teleoperation while enabling geometric understanding. Synthetic grasp generation is promising because it can produce large datasets of grasp poses through an optimization process guided by energy functions that approximate force closure, along with evaluation pipelines in simulation. These datasets are often used to train diffusion-based grasp samplers. However, a significant portion of the generated grasps are physically implausible or unnatural. Because grasp proposals are initialized by sampling points around the object's convex hull, many grasps target physically inaccessible or unsuitable regions.

Moreover, downstream manipulation tasks require the hand to interact with specific, task-relevant regions of objects, such as a handle or button. Existing synthetic grasping pipelines generate grasps indiscriminately over the object surface, leading to datasets that are poorly aligned with the needs of task-conditioned manipulation.

Our approach addresses this gap by combining the high-level VLM-based semantic priors with physically grounded, task-aware synthetic grasp generation. To this end, we propose a pipeline that first translates semantic input specifying a task into predictions of useful regions on objects using a

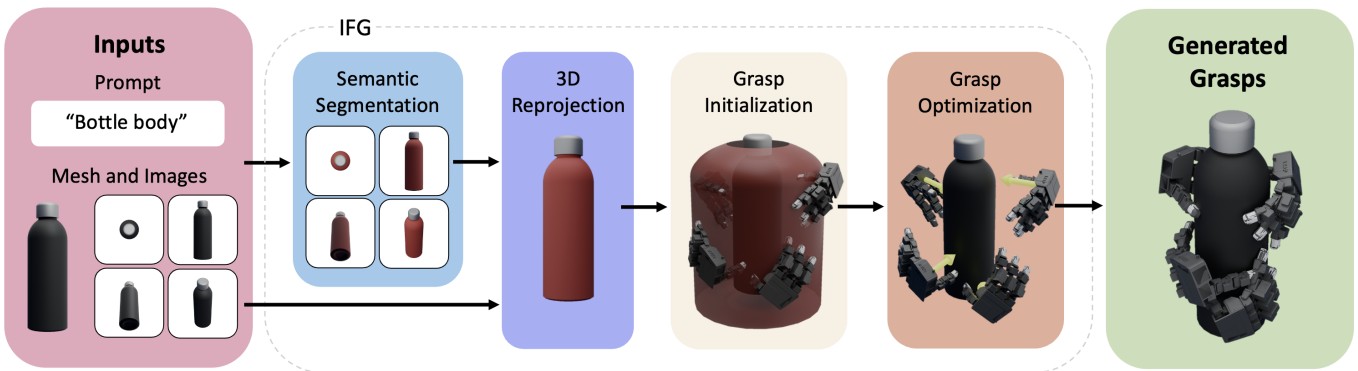

Fig. 2: IFG takes an object mesh and a task prompt as input. To incorporate semantic understanding, it renders the object from multiple viewpoints, applies a VLM-based segmentation model combining SAM[1] and VLPart[2], and reprojects the results into 3D space to identify task-relevant regions. For geometric grounding, it initializes a force closure objective at these regions and optimizes for functional grasps. The resulting data is then used to train a diffusion model for fast grasp synthesis from depth.

VLM. Then, we seed the grasp generation process on this prior to enable semantic-guided grasp synthesis, producing stable, natural grasps aligned with the demands of the task. Our pipeline is highly parallelizable, efficient, and compatible with arbitrary objects, scenes (including cluttered environments), and dexterous hands. This pipeline generates semantically meaningful grasps without any teleoperation or video data.

## II. RELATED WORK

**Dexterous Grasping Generation in Simulation.** Prior work generates dexterous grasps in simulation by adapting eigen-grasps [4, 5] or optimizing through differentiable simulation [6, 7, 8, 9] and force-closure objectives [10, 11]. We build on the force-closure formulation used in DexGraspNet but seed initialization with semantic regions rather than sampling the full object surface. Recent concurrent work also explores using VLMs for semantic grasp-region proposal [12, 13, 14]. Generated grasps are often distilled into neural network models for faster inference using VAEs [15, 11, 16, 17, 18, 3], diffusion models [19, 20], and point-cloud conditioning [21, 22, 23], with some approaches further improving test-time performance through adaptation [24, 16].

**Vision-based Dexterous Grasping.** Another line of work learns dexterous policies from real-world human interactions. Some datasets provide accurate motion-capture supervision but remain limited in scale[25, 26, 27, 28], while larger egocentric datasets require pose reconstruction [29, 30, 31]. Other approaches leverage human activity data by extracting cost functions [32, 33] or mapping human actions to robots through aligned [34, 35] or unaligned demonstrations [36, 37]. In contrast, our method avoids manual human data collection entirely by generating task-aware grasps directly in simulation.

**VLMs for Robotic Grasping.** Recent works integrate large-scale vision language models with grasping to extract affordances for 3D value maps [38], which are combined with motion planners to synthesize trajectories in a zero-shot manner [38, 39] or train downstream policies [40]. Other approaches generate vision-language-action (VLA) represen-tations or language-based plans to execute[41, 42, 43]. Most

focus on parallel-jaw grippers or high-level planning, leaving low-level dexterous grasp synthesis to motion planners or downstream policies. Our work instead uses VLMs to identify task-relevant regions that directly guide low-level dexterous grasp generation.

## III. METHOD

The goal of IFG is to efficiently generate a large grasping dataset with geometrically accurate and semantically meaning-ful grasps of a robotic hand and distill it into a general-purpose model that predicts feasible grasps in a scene. A dexterous grasp is defined as $g = (T, R, \theta)$, where $T \in \mathbb{R}^3$ and $R \in SO(3)$ denote wrist translation and rotation, and $\theta \in \mathbb{R}^{\text{DoF}}$ denotes hand joint angles (DoF = 16 for LEAP Hand [44]).

### A. Useful Region Proposal

IFG leverages knowledge from a VLM $f$ to identify objects of interest and part-level regions for grasping, which are called useful regions. To extract 2D semantic knowledge to 3D scenes, we capture $n$ RGB images from angles uniformly sampled on a camera initialization surface $S$. For single objects, $S$ is spherical, while for cluttered scenes it is a dome to reduce occlusion. A VLM $f$ is prompted to produce semantic labels, which guide a language-conditioned segmentation model (SAM [1]) and a part-level model (VLPart [2]) to produce segmentation masks of useful regions. The resulting 2D masks are deprojected to 3D points on the object mesh. To account for occlusion errors, we filter points using a two-means clustering process based on segmentation mask size. Valid deprojected points are mapped to the closest mesh faces. A voting algorithm then selects the top 60% of faces as the useful region $U$.

### B. Geometric Grasp Synthesis

We compute the segmented convex hull of the object to include only faces projected from $U$. For each grasp, the hand is initialized on the inflated convex hull by farthest point sampling with noise added to wrist pose and joint angles. An optimization process performs gradient descent against an

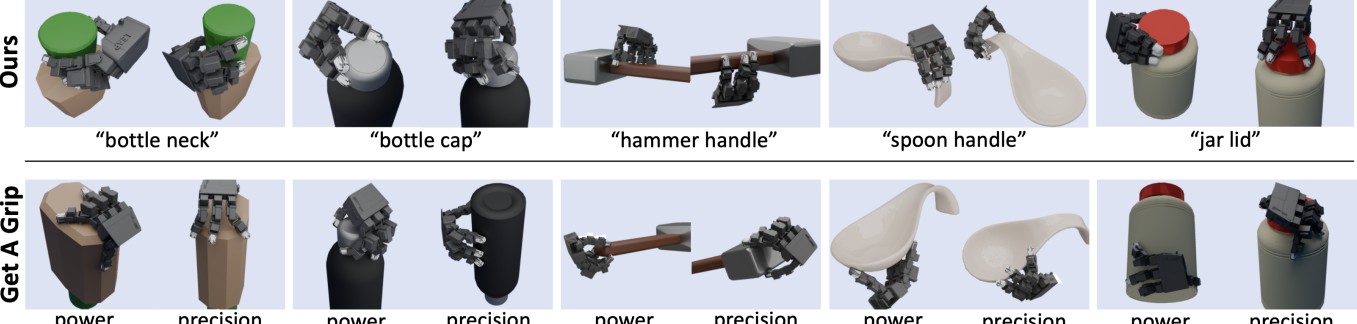

Fig. 3: Compared to Get a Grip's synthetic grasp generation method, our method produces more human-like grasps. For instance, Get a Grip often grasps the bottom of the bottle, while our method knows to robustly grasp the neck. Please see our website for 3D visualizations.

energy term

$$E = E_{\text{fc}} + w_{\text{dis}}E_{\text{dis}} + w_{\text{joints}}E_{\text{joints}} + w_{\text{pen}}E_{\text{pen}} + w_{\text{spen}}E_{\text{spen}}$$

where $E_{\text{fc}}$ approximates force closure of the grasp, $E_{\text{dis}}$ encourages hand-object proximity, based on the contact points of the hand, $E_{\text{joints}}$, $E_{\text{pen}}$, and $E_{\text{spen}}$ respectively penalizes joint violations, hand-object penetration, and self-penetration of the hand. For the single object setting, we exclude the tabletop by setting $w_{\text{spen}} = 0$ to produce more diverse grasps. Relative to Get a Grip, we make two key changes: replace precision grasps with power grasps that utilize the inside regions of all fingers instead of fingertip-only and initialize on the segmented convex hull rather than the full hull. These changes improve stability and functional alignment.

### C. Simulation Evaluation

To ensure the robustness of generated grasps, we perform tasks with them in a simulation environment. Each evaluation proceeds in three phases: (1) the grasp and object are initialized in a simulation environment, (2) fingers are closed to secure the object, and (3) task execution is performed. Following Get a Grip, we assign each grasp a smooth label by perturbing its joint angles to generate $d$ additional grasps, evaluating all $d+1$

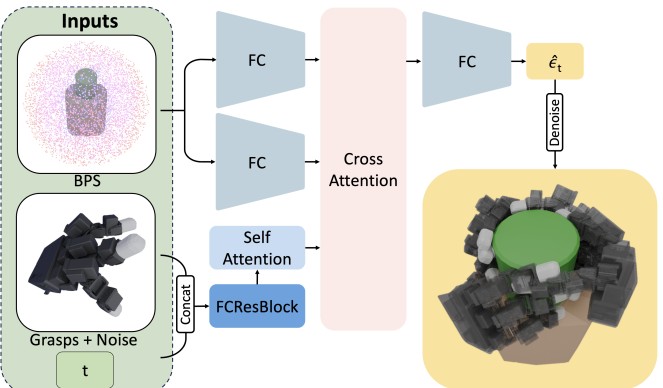

Fig. 4: The generated grasp data is distilled into a diffusion model that only expects proprioceptive observation inputs obtainable from a single depth camera. The model converts depth camera data into a Basis Point Set (BPS) and refines randomly sampled noise into functional grasps on semantically meaningful regions of the object. The architecture of the diffusion model follows a similar design to Get a Grip [3].

variants, and averaging their binary success outcomes. Grasps with low smooth success are discarded, yielding a dataset $G$ of robust force-closure power grasps. In our experiments, $d = 5$.

### D. Diffusion Model Distillation

While the grasping pipeline produces diverse and semantically meaningful candidate grasps, it is too slow for direct inference and depends on privileged mesh information. Hence, we distill the grasps into a generative neural network. We generate 2.5 million grasps using our pipeline on DexGrasp-Net2's scene dataset. Based on the end effector position, partial pointcloud observations are captured for each grasp to simulate a depth camera input, which are converted to basis point sets (BPS), a structured point-cloud representation [45] shown in [3] to be a better representation to train the model on. The resulting dataset is used to train a diffusion model that generates grasps from basis point set input. The resulting model inherits both the geometric reasoning capabilities of the training pipeline and the semantic understanding provided by the vision-language model (VLM), as illustrated in Figure 4.

| Grasp Optimization Pipeline | Single (%) | Cluttered (%) |
|---|---|---|
| *Ours* | | |
| single camera only | 47.83 | 18.53 |
| + multi-camera around object | 48.37 | 24.70 |
| + two-means clustering | 49.04 | 31.59 |
| + voting-based filtering (full pipeline) | **51.11** | **32.23** |
| *Reference baseline* | 50.93 | 14.58 |

TABLE I: Incremental improvements from our synthetic grasp generation pipeline in single-object and cluttered-scene settings in terms of success rate under the Lift metric. For single-object evaluation, the reference baseline is Get a Grip's synthetic pipeline; for cluttered scenes, the reference baseline is DexGraspNet2's synthetic pipeline.

### IV. EXPERIMENTAL SETUP

Datasets of grasps are generated on diverse objects in both single-object and clustered-scene settings, followed by extensive simulation to evaluate robustness. The evaluation addresses three key questions: (1) Can robust and stable grasps be produced on individual objects? (2) In clustered scenes, can the object of interest be identified and grasped without collision? (3) Do the resulting grasps exhibit natural, human-like qualities suitable for functional manipulation?

| Object | DEXGRASPNET2 | GRASPTTA | ISAGRASP | OURS |
|---|---|---|---|---|
| *Selected Individual Objects* | | | | |
| Tomato Soup Can | 47.8 | 38.3 | **52.0** | 45.5 |
| Mug | 33.2 | 26.9 | 22.6 | **60.4** |
| Drill | 32.1 | 20.8 | 36.4 | **57.5** |
| Scissors | 9.7 | 0.0 | **33.7** | 20.2 |
| Screw Driver | 0.0 | 8.3 | **40.0** | 22.0 |
| Shampoo Bottle | 50.6 | 25.4 | 18.8 | **53.1** |
| Elephant Figure | 23.6 | 29.6 | 24.2 | **35.8** |
| Peach Can | **61.8** | 28.0 | 55.3 | 60.3 |
| Face Cream Tube | 32.1 | 22.5 | 20.7 | **35.5** |
| Tape Roll | 22.7 | 13.9 | 9.8 | **43.2** |
| Camel Toy | 12.8 | 14.3 | 21.3 | **21.8** |
| Body Wash | 40.2 | 22.3 | 29.4 | **58.3** |
| **Object Average** | 30.55 | 20.86 | 30.35 | **42.80** |
| **Scene Average** | **36.71** | 25.64 | 32.51 | 34.16 |

TABLE II: Trained grasp generation model success rates for crowded-scene evaluation on the lift task. Both success rates of grasps on selected challenging objects and of all grasps across all scenes are reported. IFG significantly outperforms baselines on difficult objects and has comparable performance on scene average success.

**Task Setup** We evaluate 24 single objects from Get a Grip at 5 scales, generating 200 grasps per method, and 35 dense cluttered scenes as a test set from DexGraspNet2 with 256 grasps per scene. All objects are drawn from common daily manipulation tasks, and all grasps are executed using the LEAP Hand [44]. Please also visit our website at https://ifgrasping. github.io/ for more visualizations of these results.

**Simulation Evaluation.** Grasps are tested in IsaacGym [46]. Single-object tasks include Lift (vertical translation) and Pick & Shake (lifting with perturbations). Success requires the object's relative pose to the palm to remain stable without interpenetration. Clustered scenes are evaluated only on Lift to avoid trivial collisions.

## V. RESULTS

### A. Grasp Generation Pipeline

Single-view VLM segmentation is prone to occlusion errors, such as hidden mug handles. Therefore, we capture images from multiple cameras in parallel and apply geometric filtering. Table I demonstrates that each additional technique improves the success rate of our synthetic pipeline, allowing our method to beat baseline synthetic methods by a significant margin in cluttered scenes.

### B. Single Object Grasping

A useful grasp is not only robust but also natural, both of which can be achieved through our method. As illustrated in Table I, IFG achieves a higher success rate over the reference baseline for the single-object setting, demonstrating that conditioning on part-level segmentation produces more robust grasps. Qualitative comparisons in Figure 3 demonstrate our grasps concentrate on functionally relevant regions, whereas baseline grasps from Get a Grip often target unhelpful regions simply because they cover a high percentage of the convex hull. For example, their grasps tend to grasp the head of a hammer since it covers a high percentage of the convex hull,

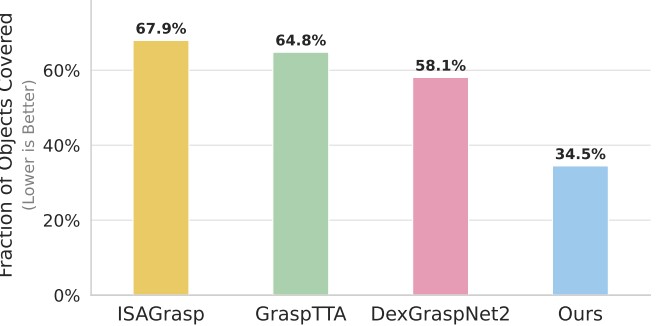

Fig. 5: When generating grasps, confidence-based methods sample a pool of grasps and select the most confident one for the scene as the final prediction. This causes most output grasp predictions to be concentrated on the easiest-to-grasp object. On the other hand, our grasp prediction model has a much more balanced coverage of the objects and a more informative success rate.

while our grasps, initialized on the segmented convex hull of the handle, are functionally correct. We hypothesize that semantic conditioning inherently improves robustness because everyday objects are designed with affordances that support secure functional grasping.

### C. Multi-object Dense Scene Grasping

Daily scenarios are often not so simple as single object settings because they involve cluttered objects requiring precise identification and firm grasps while avoiding collision. We evaluated against recent baselines [11, 18, 5] on 35 dense test scenes with 256 samples per scene. Shown in Table II, our model achieves a highly competitive global scene average success rate. Figure 1 shows our grasps of four scenes. The baselines' confidence-based sampling biases the model heavily toward easy, peripheral objects. As shown in Figure 5, the baseline concentrates output on the easiest targets and rarely proposes grasps on harder objects. In contrast, our method avoids this overfitting by not favoring particular objects in a scene. Therefore, when evaluated on a subset of challenging objects across scenes, our model significantly outperforms the baselines, demonstrating better generalization across objects. DexGraspNet2 [11] reports higher overall success rates in their paper on a relaxed 3cm lift threshold, while ours is 20cm.

## VI. CONCLUSION AND LIMITATIONS

We introduced IFG, a massively parallelizable pipeline combining internet-scale VLM semantics with geometric force-closure optimization to generate robust, functional grasps in cluttered environments without manual data collection. Distilled into a diffusion model, it infers generalized grasps from depth input. Nonetheless, our work has limitations. Currently, our method is limited to static image segmentation and force-closure grasps; extending this to continuous video streaming for dynamic manipulation remains an exciting area for future work.

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
