# OpenReview forum: "IFG: Internet-Scale Guidance for Functional Grasping Generation"
_IEEE.org/ICRA/2026/Workshop/Manipulation_Robustness — ICRA 2026_

### Official Review · Reviewer_6aU3 · 2026-05-03
**Promising semantic guidance for dexterous grasp generation, with limited functional and real-world validation**

**Rating:** 6
**Confidence:** 5

**Review:**

This paper presents IFG, a synthetic dexterous grasp-generation pipeline that uses VLM-based part segmentation to guide where force-closure grasp optimization should be initialized. Rather than sampling grasps over the entire object surface, the method biases grasp generation toward task-relevant regions such as handles, caps, necks, and lids, which is a meaningful direction for making synthetic grasp datasets more aligned with functional manipulation. The pipeline is technically coherent: multi-view renderings are segmented with SAM/VLPart, projected back to 3D mesh faces, filtered through clustering and voting, and then used to initialize force-closure optimization on a segmented convex hull. I also find the shift from fingertip-only precision grasps to power grasps using the inner regions of the fingers reasonable, since it better matches robust object-level grasping with the LEAP Hand.

The simulation results support the main direction of the paper: the ablations show that multi-view perception and filtering improve cluttered-scene grasp generation, and the learned diffusion model performs well on several challenging objects where semantic part selection is important. However, the current simulation evaluation is still limited: Lift and Pick-and-Shake mainly measure whether the object remains grasped, but they do not show whether the selected grasp enables downstream functions such as twisting, pressing, pouring, or tool use. The synthetic pipeline also relies on privileged mesh-level information and idealized simulation assumptions during data generation and evaluation, so it is not clear how robust the learned model would be under realistic depth noise, segmentation errors, pose uncertainty, friction variation, and execution errors. In addition, the paper does not include real-robot experiments, which further weakens the claim of real-world robustness. The scene-level results also need a more careful discussion, since the method improves selected challenging-object performance but is not clearly better than DexGraspNet2 on overall scene average. Another important limitation is that the useful-region proposal itself is not analyzed deeply enough; prompt ambiguity, segmentation errors, or occlusion could still lead the optimizer to produce stable but semantically wrong grasps. Overall, this is a solid and relevant workshop paper with a clear idea and promising simulation results, but the authors should moderate the functional/real-world claims and add stronger analysis of semantic-region failures, downstream functional tasks, and sim-to-real robustness.

---

### Decision · Program_Chairs · 2026-05-21

Accept